# Principal Geodesic Analysis for Probability Measures under the Optimal Transport Metric

**Vivien Seguy**
Graduate School of Informatics
Kyoto University
vivien.seguy@iip.ist.i.kyoto-u.ac.jp

**Marco Cuturi**
Graduate School of Informatics
Kyoto University
mcuturi@i.kyoto-u.ac.jp

## Abstract

Given a family of probability measures in $P(\mathcal{X})$, the space of probability measures on a Hilbert space $\mathcal{X}$, our goal in this paper is to highlight one ore more curves in $P(\mathcal{X})$ that summarize efficiently that family. We propose to study this problem under the optimal transport (Wasserstein) geometry, using curves that are restricted to be geodesic segments under that metric. We show that concepts that play a key role in Euclidean PCA, such as data centering or orthogonality of principal directions, find a natural equivalent in the optimal transport geometry, using Wasserstein means and differential geometry. The implementation of these ideas is, however, computationally challenging. To achieve scalable algorithms that can handle thousands of measures, we propose to use a relaxed definition for geodesics and regularized optimal transport distances. The interest of our approach is demonstrated on images seen either as shapes or color histograms.

## 1 Introduction

Optimal transport distances (Villani, 2008), *a.k.a* Wasserstein or earth mover's distances, define a powerful geometry to compare probability measures supported on a metric space $\mathcal{X}$. The Wasserstein space $P(\mathcal{X})$—the space of probability measures on $\mathcal{X}$ endowed with the Wasserstein distance—is a metric space which has received ample interest from a theoretical perspective. Given the prominent role played by probability measures and feature histograms in machine learning, the properties of $P(\mathcal{X})$ can also have practical implications in data science. This was shown by Agueh and Carlier (2011) who described first Wasserstein *means* of probability measures. Wasserstein means have been recently used in Bayesian inference (Srivastava et al., 2015), clustering (Cuturi and Doucet, 2014), graphics (Solomon et al., 2015) or brain imaging (Gramfort et al., 2015). When $\mathcal{X}$ is not just metric but also a Hilbert space, $P(\mathcal{X})$ is an infinite-dimensional Riemannian manifold (Ambrosio et al. 2006, Chap. 8; Villani 2008, Part II). Three recent contributions by Boissard et al. (2015, §5.2), Bigot et al. (2015) and Wang et al. (2013) exploit directly or indirectly this structure to extend Principal Component Analysis (PCA) to $P(\mathcal{X})$. These important seminal papers are, however, limited in their applicability and/or the type of curves they output. Our goal in this paper is to propose more general and scalable algorithms to carry out Wasserstein principal geodesic analysis on probability measures, and not simply dimensionality reduction as explained below.

**Principal Geodesics in $P(\mathcal{X})$ *vs.* Dimensionality Reduction on $P(\mathcal{X})$** We provide in Fig. 1 a simple example that illustrates the motivation of this paper, and which also shows how our approach differentiates itself from existing dimensionality reduction algorithms (linear and non-linear) that draw inspiration from PCA. As shown in Fig. 1, linear PCA cannot produce components that remain in $P(\mathcal{X})$. Even more advanced tools, such as those proposed by Hastie and Stuetzle (1989), fall slightly short of that goal. On the other hand, Wasserstein geodesic analysis yields geodesic components in $P(\mathcal{X})$ that are easy to interpret and which can also be used to reduce dimensionality.

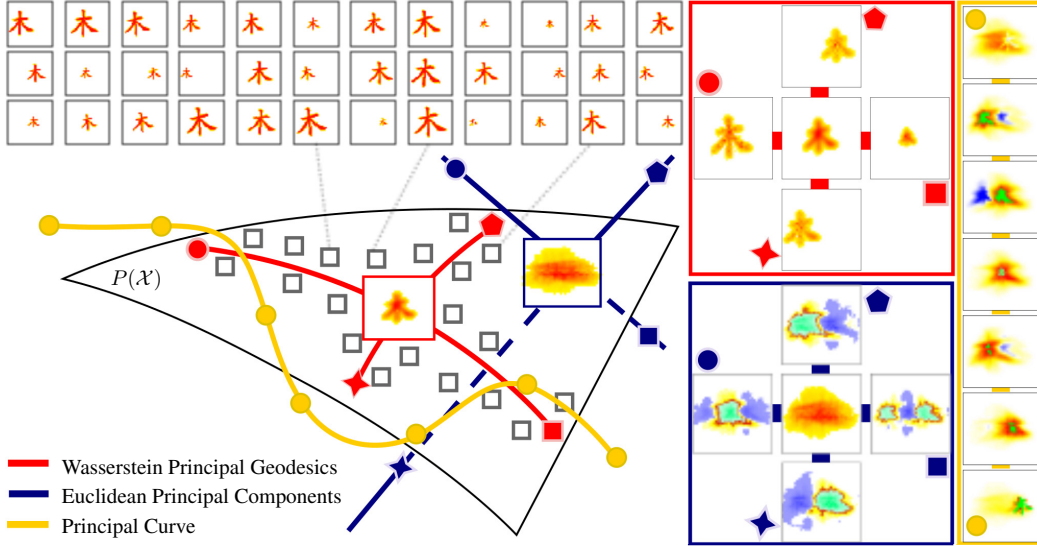

Figure 1: (top-left) Dataset: $60 \times 60$ images of a single Chinese character randomly translated, scaled and slightly rotated (36 images displayed out of 300 used). Each image is handled as a normalized histogram of $3,600$ non-negative intensities. (middle-left) Dataset schematically drawn on $P(\mathcal{X})$. The Wasserstein principal geodesics of this dataset are depicted in red, its Euclidean components in blue, and its principal curve (Verbeek et al., 2002) in yellow. (right) Actual curves (blue colors depict negative intensities, green intensities $\geq 1$). Neither the Euclidean components nor the principal curve belong to $P(\mathcal{X})$, nor can they be interpreted as meaningful axis of variation.

**Foundations of PCA and Riemannian Extensions**  Carrying out PCA on a family $(x_1, \ldots, x_n)$ of points taken in a space $X$ can be described in abstract terms as: *(i)* define a mean element $\bar{x}$ for that dataset; *(ii)* define a family of *components* in $X$, typically geodesic curves, that contain $\bar{x}$; *(iii)* fit a component by making it follow the $x_i$'s as closely as possible, in the sense that the sum of the distances of each point $x_i$ to that component is minimized; *(iv)* fit additional components by iterating step *(iii)* several times, with the added constraint that each new component is different (orthogonal) enough to the previous components. When $X$ is Euclidean and the $x_i$'s are vectors in $\mathbb{R}^d$, the $(n+1)$-th component $v_{n+1}$ can be computed iteratively by solving:

$$v_{n+1} \in \underset{v \in V_n^{\perp}, \|v\|_2 = 1}{\operatorname{argmin}} \sum_{i=1}^{N} \min_{t \in \mathbb{R}} \|x_i - (\bar{x} + tv)\|_2^2, \text{ where } V_0 \overset{\text{def.}}{=} \emptyset, \text{ and } V_n \overset{\text{def.}}{=} \operatorname{span}\{v_1, \cdots, v_n\}. \quad (1)$$

Since PCA is known to boil down to a simple eigen-decomposition when $X$ is Euclidean or Hilbertian (Schölkopf et al., 1997), Eq. (1) looks artificially complicated. This formulation is, however, extremely useful to generalize PCA to Riemannian manifolds (Fletcher et al., 2004). This generalization proceeds first by replacing vector means, lines and orthogonality conditions using respectively Fréchet means (1948), geodesics, and orthogonality in tangent spaces. Riemannian PCA builds then upon the knowledge of the *exponential map* at each point $x$ of the manifold $X$. Each exponential map $\exp_x$ is locally bijective between the tangent space $T_x$ of $x$ and $X$. After computing the Fréchet mean $\bar{x}$ of the dataset, the logarithmic map $\log_{\bar{x}}$ at $\bar{x}$ (the inverse of $\exp_{\bar{x}}$) is used to map all data points $x_i$ onto $T_{\bar{x}}$. Because $T_{\bar{x}}$ is a Euclidean space by definition of Riemannian manifolds, the dataset $(\log_{\bar{x}} x_i)_i$ can be studied using Euclidean PCA. Principal geodesics in $X$ can then be recovered by applying the exponential map to a principal component $v^{\star}$, $\{\exp_{\bar{x}}(tv^{\star}), |t| < \varepsilon\}$.

**From Riemannian PCA to Wasserstein PCA: Related Work**  As remarked by Bigot et al. (2015), Fletcher et al.'s approach cannot be used as it is to define Wasserstein geodesic PCA, because $P(\mathcal{X})$ is infinite dimensional and because there are no known ways to define exponential maps which are locally bijective between Wasserstein tangent spaces and the manifold of probability measures. To circumvent this problem, Boissard et al. (2015), Bigot et al. (2015) have proposed to formulate the geodesic PCA problem directly as an optimization problem over curves in $P(\mathcal{X})$.

Boissard et al. and Bigot et al. study the Wasserstein PCA problem in restricted scenarios: Bigot et al. focus their attention on measures supported on $\mathcal{X} = \mathbb{R}$, which considerably simplifies their analysis since it is known in that case that the Wasserstein space $P(\mathbb{R})$ can be embedded isometrically in $L^1(\mathbb{R})$; Boissard et al. assume that each input measure has been generated from a single template density (the mean measure) which has been transformed according to one "admissible deformation" taken in a parameterized family of deformation maps. Their approach to Wasserstein PCA boils down to a functional PCA on such maps. Wang et al. proposed a more general approach: given a family of input empirical measures $(\mu_1, \ldots, \mu_N)$, they propose to compute first a "template measure" $\tilde{\mu}$ using $k$-means clustering on $\sum_i \mu_i$. They consider next all optimal transport plans $\pi_i$ between that template $\tilde{\mu}$ and each of the measures $\mu_i$, and propose to compute the barycentric projection (see Eq. 8) of each optimal transport plan $\pi_i$ to recover Monge maps $T_i$, on which standard PCA can be used. This approach is computationally attractive since it requires the computation of only one optimal transport per input measure. Its weakness lies, however, in the fact that the curves in $P(\mathcal{X})$ obtained by displacing $\tilde{\mu}$ along each of these PCA directions are not geodesics in general.

**Contributions and Outline** We propose a new algorithm to compute Wasserstein Principal Geodesics (WPG) in $P(\mathcal{X})$ for arbitrary Hilbert spaces $\mathcal{X}$. We use several approximations—both of the optimal transport metric and of its geodesics—to obtain tractable algorithms that can scale to thousands of measures. We provide first in §2 a review of the key concepts used in this paper, namely Wasserstein distances and means, geodesics and tangent spaces in the Wasserstein space. We propose in §3 to parameterize a Wasserstein principal component (PC) using two velocity fields defined on the support of the Wasserstein mean of all measures, and formulate the WPG problem as that of optimizing these velocity fields so that the average distance of all measures to that PC is minimal. This problem is non-convex and non-smooth. We propose to optimize smooth upper-bounds of that objective using entropy regularized optimal transport in §4. The practical interest of our approach is demonstrated in §5 on toy samples, datasets of shapes and histograms of colors.

**Notations** We write $\langle A, B \rangle$ for the Frobenius dot-product of matrices $A$ and $B$. $\mathbf{D}(u)$ is the diagonal matrix of vector $u$. For a mapping $f : \mathcal{Y} \to \mathcal{Y}$, we say that $f$ acts on a measure $\mu \in P(\mathcal{Y})$ through the pushforward operator # to define a new measure $f \# \mu \in P(\mathcal{Y})$. This measure is characterized by the identity $(f \# \mu)(B) = \mu(f^{-1}(B))$ for any Borel set $B \subset \mathcal{Y}$. We write $p_1$ and $p_2$ for the canonical projection operators $\mathcal{X}^2 \to \mathcal{X}$, defined as $p_1(x_1, x_2) = x_1$ and $p_2(x_1, x_2) = x_2$.

## 2 Background on Optimal Transport

**Wasserstein Distances** We start this section with the main mathematical object of this paper:

**Definition 1.** *(Villani, 2008, Def. 6.1) Let $P(\mathcal{X})$ the space of probability measures on a Hilbert space $\mathcal{X}$. Let $\Pi(\nu, \eta)$ be the set of probability measures on $\mathcal{X}^2$ with marginals $\nu$ and $\eta$, i.e. $p_1 \# \pi = \nu$ and $p_2 \# \pi = \eta$. The squared 2-Wasserstein distance between $\nu$ and $\eta$ in $P(\mathcal{X})$ is defined as:*

$$W_2^2(\nu, \eta) = \inf_{\pi \in \Pi(\nu, \eta)} \int_{\mathcal{X}^2} \|x - y\|_{\mathcal{X}}^2 d\pi(x, y). \tag{2}$$

**Wasserstein Barycenters** Given a family of $N$ probability measures $(\mu_1, \cdots, \mu_N)$ in $P(\mathcal{X})$ and weights $\lambda \in \mathbb{R}_+^N$, Agueh and Carlier (2011) define $\bar{\mu}$, the Wasserstein barycenter of these measures:

$$\bar{\mu} \in \operatorname*{argmin}_{\nu \in P(\mathcal{X})} \sum_{i=1}^{N} \lambda_i W_2^2(\mu_i, \nu).$$

Our paper relies on several algorithms which have been recently proposed (Benamou et al., 2015; Bonneel et al., 2015; Carlier et al., 2015; Cuturi and Doucet, 2014) to compute such barycenters.

**Wasserstein Geodesics** Given two measures $\nu$ and $\eta$, let $\Pi^\star(\nu, \eta)$ be the set of optimal couplings for Eq. (2). Informally speaking, it is well known that if either $\nu$ or $\eta$ are absolutely continuous measures, then any optimal coupling $\pi^\star \in \Pi^\star(\nu, \eta)$ is degenerated in the sense that, assuming for instance that $\nu$ is absolutely continuous, for all $x$ in the support of $\nu$ only one point $y \in \mathcal{X}$ is such that $d\pi^\star(x, y) > 0$. In that case, the optimal transport is said to have *no* mass splitting, and

there exists an optimal mapping $T : \mathcal{X} \to \mathcal{X}$ such that $\pi^\star$ can be written, using a pushforward, as $\pi^\star = (\mathrm{id} \times T)\#\nu$. When there is no mass splitting to transport $\nu$ to $\eta$, McCann's interpolant (1997):

$$g_t = ((1 - t)\mathrm{id} + tT)\#\nu, \ \ t \in [0, 1], \tag{3}$$

defines a geodesic curve in the Wasserstein space, *i.e.* $(g_t)_t$ is locally the shortest path between any two measures located on the geodesic, with respect to $W_2$. In the more general case, where no optimal map $T$ exists and mass splitting occurs (for some locations $x$ one may have $d\pi^\star(x, y) > 0$ for several $y$), then a geodesic can still be defined, but it relies on the optimal plan $\pi^\star$ instead: $g_t = ((1 - t)p_1 + tp_2)\#\pi^\star, t \in [0, 1]$, (Ambrosio et al., 2006, §7.2). Both cases are shown in Fig. 2.

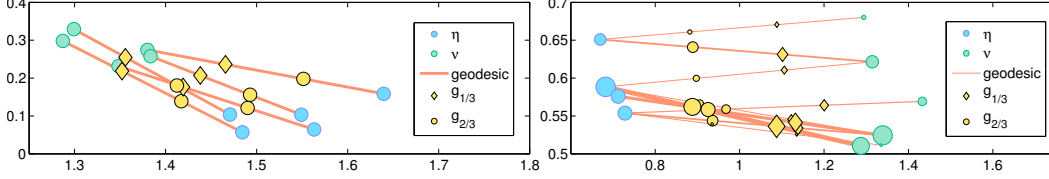

Figure 2: Both plots display geodesic curves between two empirical measures $\nu$ and $\eta$ on $\mathbb{R}^2$. An optimal map exists in the left plot (no mass splitting occurs), whereas some of the mass of $\nu$ needs to be split to be transported onto $\eta$ on the right plot.

**Tangent Space and Tangent Vectors** We briefly describe in this section the tangent spaces of $P(\mathcal{X})$, and refer to (Ambrosio et al., 2006, Chap. 8) for more details. Let $\mu : I \subset \mathbb{R} \to P(\mathcal{X})$ be a curve in $P(\mathcal{X})$. For a given time $t$, the tangent space of $P(\mathcal{X})$ at $\mu_t$ is a subset of $L^2(\mu_t, \mathcal{X})$, the space of square-integrable velocity fields supported on $\mathrm{Supp}(\mu_t)$. At any $t$, there exists tangent vectors $v_t$ in $L^2(\mu_t, \mathcal{X})$ such that $\lim_{h \to 0} W_2(\mu_{t+h}, (\mathrm{id} + hv_t)\#\mu_t)/|h| = 0$. Given a geodesic curve in $P(\mathcal{X})$ parameterized as Eq. (3), its corresponding tangent vector at time zero is $v = T - \mathrm{id}$.

## 3  Wasserstein Principal Geodesics

**Geodesic Parameterization** The goal of principal geodesic analysis is to define geodesic curves in $P(\mathcal{X})$ that go through the mean $\bar{\mu}$ and which pass close enough to all target measures $\mu_i$. To that end, geodesic curves can be parameterized with two end points $\nu$ and $\eta$. However, to avoid dealing with the constraint that a principal geodesic needs to go through $\bar{\mu}$, one can start instead from $\bar{\mu}$, and consider a velocity field $v \in L^2(\bar{\mu}, \mathcal{X})$ which displaces all of the mass of $\bar{\mu}$ in both directions:

$$g_t(v) \stackrel{\mathrm{def.}}{=} (\mathrm{id} + tv)\#\bar{\mu}, \ \ t \in [-1, 1]. \tag{4}$$

Lemma 7.2.1 of Ambrosio et al. (2006) implies that any geodesic going through $\bar{\mu}$ can be written as Eq. (4). Hence, we do not lose any generality using this parameterization. However, given an arbitrary vector field $v$, the curve $(g_t(v))_t$ is not necessarily a geodesic. Indeed, the maps $\mathrm{id} \pm v$ are not necessarily in the set $\mathcal{C}_{\bar{\mu}} \stackrel{\mathrm{def.}}{=} \{r \in L^2(\bar{\mu}, \mathcal{X}) | (\mathrm{id} \times r)\#\bar{\mu} \in \Pi^\star(\bar{\mu}, r\#\bar{\mu})\}$ of maps that are optimal when moving mass away from $\bar{\mu}$. Ensuring thus, at each step of our algorithm, that $v$ is still such that $(g_t(v))_t$ is a geodesic curve is particularly challenging. To relax this strong assumption, we propose to use a generalized formulation of geodesics, which builds upon not one but *two* velocity fields, as introduced by Ambrosio et al. (2006, §9.2):

**Definition 2.** *(adapted from (Ambrosio et al., 2006, §9.2)) Let $\sigma$, $\nu$, $\eta \in P(\mathcal{X})$, and assume there is an optimal mapping $T^{(\sigma, \nu)}$ from $\sigma$ to $\nu$ and an optimal mapping $T^{(\sigma, \eta)}$ from $\sigma$ to $\eta$. A generalized geodesic, illustrated in Fig. 3 between $\nu$ and $\eta$ with base $\sigma$ is defined by,*

$$g_t = \left( (1 - t)T^{(\sigma, \nu)} + tT^{(\sigma, \eta)} \right)\#\sigma, \ \ t \in [0, 1].$$

Choosing $\bar{\mu}$ as the base measure in Definition 2, and two fields $v_1, v_2$ such that $\mathrm{id} - v_1, \mathrm{id} + v_2$ are optimal mappings (in $\mathcal{C}_{\bar{\mu}}$), we can define the following generalized geodesic $g_t(v_1, v_2)$:

$$g_t(v_1, v_2) \stackrel{\mathrm{def.}}{=} (\mathrm{id} - v_1 + t(v_1 + v_2))\#\bar{\mu}, \ \text{for } t \in [0, 1]. \tag{5}$$

Generalized geodesics become true geodesics when $v_1$ and $v_2$ are positively proportional. We can thus consider a regularizer that controls the deviation from that property by defining $\Omega(v_1, v_2) = (\langle v_1, v_2 \rangle_{L^2(\bar{\mu}, \mathcal{X})} - \|v_1\|_{L^2(\bar{\mu}, \mathcal{X})} \|v_2\|_{L^2(\bar{\mu}, \mathcal{X})})^2$, which is minimal when $v_1$ and $v_2$ are indeed positively proportional. We can now formulate the WPG problem as computing, for $n \geq 0$, the $(n+1)^{\text{th}}$ principal (generalized) geodesic component of a family of measures $(\mu_i)_i$ by solving, with $\lambda > 0$:

$$\min_{v_1, v_2 \in L^2(\bar{\mu}, \mathcal{X})} \lambda \Omega(v_1, v_2) + \sum_{i=1}^{N} \min_{t \in [0,1]} W_2^2 \left(g_t(v_1, v_2), \mu_i\right), \text{s.t.} \begin{cases} \text{id} - v_1, \text{id} + v_2 \in \mathcal{C}_{\bar{\mu}}, \\ v_1 + v_2 \in \text{span}(\{v_1^{(i)} + v_2^{(i)}\}_{i \leq n})^{\perp}. \end{cases}$$

(6)

This problem is not convex in $v_1$, $v_2$. We propose to find an approximation of that minimum by a projected gradient descent, with a projection that is to be understood in terms of an alternative metric on the space of vector fields $L^2(\bar{\mu}, \mathcal{X})$. To preserve the optimality of the mappings $\text{id} - v_1$ and $\text{id} + v_2$ between iterations, we introduce in the next paragraph a suitable projection operator on $L^2(\bar{\mu}, \mathcal{X})$.

**Remark 1.** *A trivial way to ensure that $(g_t(v))_t$ is geodesic is to impose that the vector field $v$ is a translation, namely that $v$ is uniformly equal to a vector $\tau$ on all of $Supp(\bar{\mu})$. One can show in that case that the WPG problem described in Eq. (6) outputs an optimal vector $\tau$ which is the Euclidean principal component of the family formed by the means of each measure $\mu_i$.*

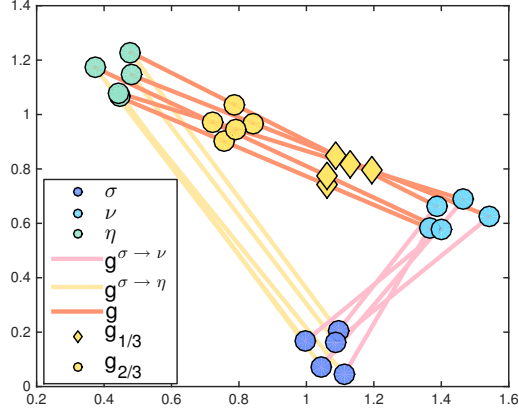

Figure 3: Generalized geodesic interpolation between two empirical measures $\nu$ and $\eta$ using the base measure $\sigma$, all defined on $\mathcal{X} = \mathbb{R}^2$.

**Projection on the Optimal Mapping Set** We use a projected gradient descent method to solve Eq. (6) approximately. We will compute the gradient of a local upper-bound of the objective of Eq. (6) and update $v_1$ and $v_2$ accordingly. We then need to ensure that $v_1$ and $v_2$ are such that $\text{id} - v_1$ and $\text{id} + v_2$ belong to the set of optimal mappings $\mathcal{C}_{\bar{\mu}}$. To do so, we would ideally want to compute the projection $r_2$ of $\text{id} + v_2$ in $\mathcal{C}_{\bar{\mu}}$

$$r_2 = \underset{r \in \mathcal{C}_{\bar{\mu}}}{\text{argmin}} \|(\text{id} + v_2) - r\|_{L^2(\bar{\mu}, \mathcal{X})}^2,$$

(7)

to update $v_2 \leftarrow r_2 - \text{id}$. Westdickenberg (2010) has shown that the set of optimal mappings $\mathcal{C}_{\bar{\mu}}$ is a convex closed cone in $L^2(\bar{\mu}, \mathcal{X})$, leading to the existence and the unicity of the solution of Eq. (7). However, there is to our knowledge no known method to compute the projection $r_2$ of $\text{id} + v_2$. There is nevertheless a well known and efficient approach to find a mapping $r_2$ in $\mathcal{C}_{\bar{\mu}}$ which is close to $\text{id} + v_2$. That approach, known as the the barycentric projection, requires to compute first an optimal coupling $\pi^\star$ between $\bar{\mu}$ and $(\text{id} + v_2)\#\bar{\mu}$, to define then a (conditional expectation) map

$$T_{\pi^\star}(x) \stackrel{\text{def.}}{=} \int_{\mathcal{X}} y d\pi^\star(y|x).$$

(8)

Ambrosio et al. (2006, Theorem 12.4.4) or Reich (2013, Lemma 3.1) have shown that $T_{\pi^\star}$ is indeed an optimal mapping between $\bar{\mu}$ and $T_{\pi^\star}\#\bar{\mu}$. We can thus set the velocity field as $v_2 \leftarrow T_{\pi^\star} - \text{id}$ to carry out an approximate projection. We show in the supplementary material that this operator can be in fact interpreted as a projection under a *pseudo*-metric $GW_{\bar{\mu}}$ on $L^2(\bar{\mu}, \mathcal{X})$.

## 4 Computing Principal Generalized Geodesics in Practice

We show in this section that when $\mathcal{X} = \mathbb{R}^d$, the steps outlined above can be implemented efficiently.

**Input Measures and Their Barycenter** Each input measure in the family $(\mu_1, \cdots, \mu_N)$ is a finite weighted sum of Diracs, described by $n_i$ points contained in a matrix $X_i$ of size $d \times n_i$, and a (non-negative) weight vector $a_i$ of dimension $n_i$ summing to 1. The Wasserstein mean of these measures is given and equal to $\bar{\mu} = \sum_{k=1}^{p} b_k \delta_{y_k}$, where the nonnegative vector $b = (b_1, \cdots, b_p)$ sums to one, and $Y = [y_1, \cdots, y_p] \in \mathbb{R}^{d \times p}$ is the matrix containing locations of $\bar{\mu}$.

**Generalized Geodesic** Two velocity vectors for each of the $p$ points in $\bar{\mu}$ are needed to parameterize a generalized geodesic. These velocity fields will be represented by two matrices $V_1 = [v_1^1, \cdots, v_p^1]$ and $V_2 = [v_1^2, \cdots, v_p^2]$ in $\mathbb{R}^{d \times p}$. Assuming that these velocity fields yield optimal mappings, the points at time $t$ of that generalized geodesic are the measures parameterized by $t$,

$$g_t(V_1, V_2) = \sum_{k=1}^{p} b_k \delta_{z_k^t}, \text{ with locations } Z_t = [z_1^t, \ldots, z_p^t] \stackrel{\text{def.}}{=} Y - V_1 + t(V_1 + V_2).$$

The squared 2-Wasserstein distance between datum $\mu_i$ and a point $g_t(V_1, V_2)$ on the geodesic is:
$$W_2^2(g_t(V_1, V_2), \mu_i) = \min_{P \in U(b, a_i)} \langle P, M_{Z_t X_i} \rangle, \tag{9}$$

where $U(b, a_i)$ is the transportation polytope $\{P \in \mathbb{R}_+^{p \times n_i}, P\mathbf{1}_{n_i} = b, P^T\mathbf{1}_p = a_i\}$, and $M_{Z_t X_i}$ stands for the $p \times n_i$ matrix of squared-Euclidean distances between the $p$ and $n_i$ column vectors of $Z_t$ and $X_i$ respectively. Writing $\mathbf{z}_t = \mathbf{D}(Z_t^T Z_t)$ and $\mathbf{x}_i = \mathbf{D}(X_i^T X_i)$, we have that

$$M_{Z_t X_i} = \mathbf{z}_t \mathbf{1}_{n_i}^T + \mathbf{1}_p \mathbf{x}_i^T - 2Z_t^T X_i \in \mathbb{R}^{p \times n_i},$$
which, by taking into account the marginal conditions on $P \in U(b, a_i)$, leads to,
$$\langle P, M_{Z_t X_i} \rangle = b^T \mathbf{z}_t + a_i^T \mathbf{x}_i - 2\langle P, Z_t^T X_i \rangle. \tag{10}$$

**1. Majorization of the Distance of each $\mu_i$ to the Principal Geodesic** Using Eq. (10), the distance between each $\mu_i$ and the PC $(g_t(V_1, V_2))_t$ can be cast as a function $f_i$ of $(V_1, V_2)$:

$$f_i(V_1, V_2) \stackrel{\text{def.}}{=} \min_{t \in [0,1]} \left( b^T \mathbf{z}_t + a_i^T \mathbf{x}_i + \min_{P \in U(b, a_i)} -2\langle P, (Y - V_1 + t(V_1 + V_2))^T X_i \rangle \right). \tag{11}$$

where we have replaced $Z_t$ above by its explicit form in $t$ to highlight that the objective above is quadratic convex plus piecewise linear concave as a function of $t$, and thus neither convex nor concave. Assume that we are given $P^\sharp$ and $t^\sharp$ that are approximate arg-minima for $f_i(V_1, V_2)$. For any $A, B$ in $\mathbb{R}^{d \times p}$, we thus have that each distance $f_i(V_1, V_2)$ appearing in Eq. (6), is such that

$$f_i(A, B) \leqslant m_i^{V_1 V_2}(A, B) \stackrel{\text{def.}}{=} \langle P^\sharp, M_{Z_{t^\sharp} X_i} \rangle. \tag{12}$$

We can thus use a *majorization-minimization* procedure (Hunter and Lange, 2000) to minimize the sum of terms $f_i$ by iteratively creating majorization functions $m_i^{V_1 V_2}$ at each iterate $(V_1, V_2)$. All functions $m_i^{V_1 V_2}$ are quadratic convex. Given that we need to ensure that these velocity fields yield optimal mappings, and that they may also need to satisfy orthogonality constraints with respect to lower-order principal components, we use gradient steps to update $V_1, V_2$, which can be recovered using (Cuturi and Doucet, 2014, §4.3) and the chain rule as:

$$\nabla_1 m_i^{V_1 V_2} = 2(t^\sharp - 1)(Z_{t^\sharp} - X_i P^{\sharp T} \mathbf{D}(b^{-1})), \quad \nabla_2 m_i^{V_1 V_2} = 2t^\sharp (Z_{t^\sharp} - X_i P^{\sharp T} \mathbf{D}(b^{-1})). \tag{13}$$

**2. Efficient Approximation of $P^\sharp$ and $t^\sharp$** As discussed above, gradients for majorization functions $m_i^{V_1 V_2}$ can be obtained using approximate minima $P^\sharp$ and $t^\sharp$ for each function $f_i$. Because the objective of Eq. (11) is not convex w.r.t. $t$, we propose to do an exhaustive 1-d grid search with $K$ values in $[0, 1]$. This approach would still require, in theory, to solve $K$ optimal transport problems to solve Eq. (11) for each of the $N$ input measures. To carry out this step efficiently, we propose to use entropy regularized transport (Cuturi, 2013), which allows for much faster computations and efficient parallelizations to recover approximately optimal transports $P^\sharp$.

**3. Projected Gradient Update** Velocity fields are updated with a gradient stepsize $\beta > 0$,

$$V_1 \leftarrow V_1 - \beta \left( \sum_{i=1}^{N} \nabla_1 m_i^{V_1 V_2} + \lambda \nabla_1 \Omega \right), \quad V_2 \leftarrow V_2 - \beta \left( \sum_{i=1}^{N} \nabla_2 m_i^{V_1 V_2} + \lambda \nabla_2 \Omega \right),$$

followed by a projection step to enforce that $V_1$ and $V_2$ lie in $\text{span}(V_1^{(1)} + V_2^{(1)}, \cdots, V_1^{(n)} + V_2^{(n)})^\perp$ in the $L^2(\bar{\mu}, \mathcal{X})$ sense when computing the $(n+1)^{\text{th}}$ PC. We finally apply the barycentric projection operator defined in the end of §3. We first need to compute two optimal transport plans,

$$P_1^\star \in \underset{P \in U(b,b)}{\operatorname{argmin}} \langle P, M_{Y(Y-V_1)} \rangle, \quad P_2^\star \in \underset{P \in U(b,b)}{\operatorname{argmin}} \langle P, M_{Y(Y+V_2)} \rangle, \tag{14}$$

to form the barycentric projections, which then yield updated velocity vectors:

$$V_1 \leftarrow - \left( (Y - V_1) P_1^{\star T} \mathbf{D}(b^{-1}) - Y \right), \quad V_2 \leftarrow (Y + V_2) P_2^{\star T} \mathbf{D}(b^{-1}) - Y. \tag{15}$$

We repeat steps 1,2,3 until convergence. Pseudo-code is given in the supplementary material.

# 5  Experiments

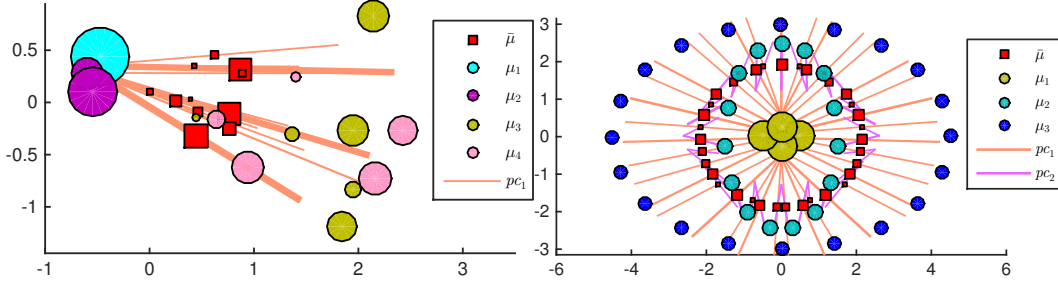

Figure 4: Wasserstein mean $\bar{\mu}$ and first PC computed on a dataset of four (left) and three (right) empirical measures. The second PC is also displayed in the right figure.

**Toy samples:** We first run our algorithm on two simple synthetic examples. We consider respectively 4 and 3 empirical measures supported on a small number of locations in $\mathcal{X} = \mathbb{R}^2$, so that we can compute their exact Wasserstein means, using the multi-marginal linear programming formulation given in (Agueh and Carlier, 2011, §4). These measures and their mean (red squares) are shown in Fig. 4. The first principal component on the left example is able to capture both the variability of average measure locations, from left to right, and also the variability in the spread of the measure locations. On the right example, the first principal component captures the overall elliptic shape of the supports of all considered measures. The second principal component reflects the variability in the parameters of each ellipse on which measures are located. The variability in the weights of each location is also captured through the Wasserstein mean, since each single line of a generalized geodesic has a corresponding location and weight in the Wasserstein mean.

**MNIST:** For each of the digits ranging from $0$ to $9$, we sample 1,000 images in the MNIST database representing that digit. Each image, originally a 28x28 grayscale image, is converted into a probability distribution on that grid by normalizing each intensity by the total intensity in the image. We compute the Wasserstein mean for each digit using the approach of Benamou et al. (2015). We then follow our approach to compute the first three principal geodesics for each digit. Geodesics for four of these digits are displayed in Fig. 5 by showing intermediary (rasterized) measures on the curves. While some deformations in these curves can be attributed to relatively simple rotations around the digit center, more interesting deformations appear in some of the curves, such as the the loop on the bottom left of digit 2. Our results are easy to interpret, unlike those obtained with Wang et al.'s approach (2013) on these datasets, see supplementary material. Fig. 6 displays the first PC obtained on a subset of MNIST composed of 2,000 images of 2 and 4 in equal proportions.

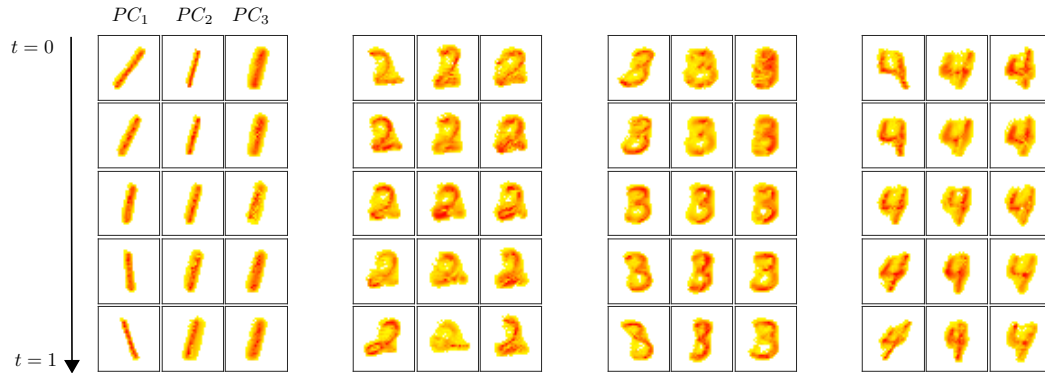

Figure 5: 1000 images for each of the digits 1,2,3,4 were sampled from the MNIST database. We display above the first three PCs sampled at times $t_k = k/4, \ k = 0, \dots, 4$ for each of these digits.

**Color histograms:** We consider a subset of the Caltech-256 Dataset composed of three image categories: waterfalls, tomatoes and tennis balls, resulting in a set of 295 color images. The pixels

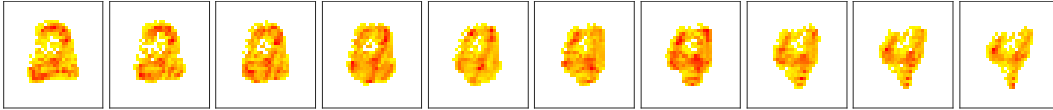

Figure 6: First PC on a subset of MNIST composed of one thousand 2s and one thousand 4s.

contained in each image can be seen as a point-cloud in the RGB color space $[0, 1]^3$. We use $k$-means quantization to reduce the size of these uniform point-clouds into a set of $k = 128$ weighted points, using cluster assignments to define the weights of each of the $k$ cluster centroids. Each image can be thus regarded as a discrete probability measure of 128 atoms in the tridimensional RGB space. We then compute the Wasserstein barycenter of these measures supported on $p = 256$ locations using (Cuturi and Doucet, 2014, Alg.2). Principal components are then computed as described in §4. The computation for a single PC is performed within 15 minutes on an iMac (3.4GHz Intel Core i7). Fig. 7 displays color palettes sampled along each of the first three PCs. The first PC suggests that the main source of color variability in the dataset is the illumination, each pixel going from dark to light. Second and third PCs display the variation of colors induced by the typical images' dominant colors (blue, red, yellow). Fig. 8 displays the second PC, along with three images projected on that curve. The projection of a given image on a PC is obtained by finding first the optimal time $t^\star$ such that the distance of that image to the PC at $t^\star$ is minimum, and then by computing an optimal color transfer (Pitié et al., 2007) between the original image and the histogram at time $t^\star$.

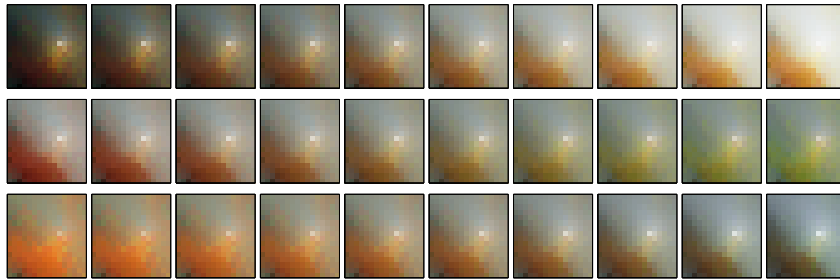

Figure 7: Each row represents a PC displayed at regular time intervals from $t = 0$ (left) to $t = 1$ (right), from the first PC (top) to the third PC (bottom).

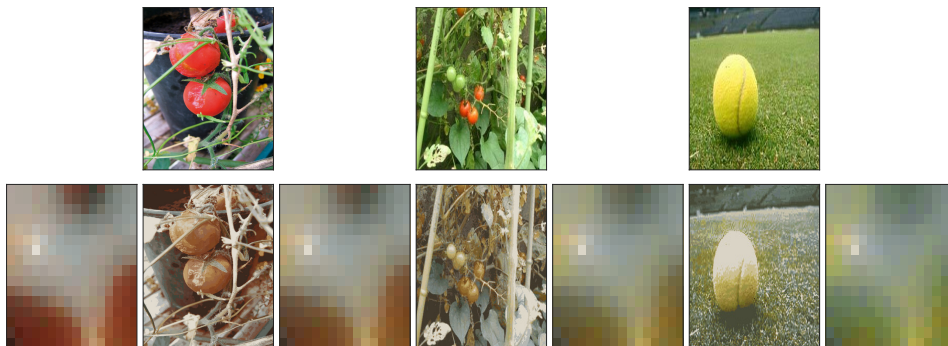

Figure 8: Color palettes from the second PC ($t = 0$ on the left, $t = 1$ on the right) displayed at times $t = 0, \frac{1}{3}, \frac{2}{3}, 1$. Images displayed in the top row are original; their projection on the PC is displayed below, using a color transfer with the palette in the PC to which they are the closest.

**Conclusion** We have proposed an approximate projected gradient descent method to compute generalized geodesic principal components for probability measures. Our experiments suggest that these principal geodesics may be useful to analyze shapes and distributions, and that they do not require any parameterization of shapes or deformations to be used in practice.

**Aknowledgements** MC acknowledges the support of JSPS young researcher A grant 26700002.

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
