[Supplementary Material]

# Principal Geodesic Analysis for Probability Measures under the Optimal Transport Metric: Supplementary Material

**Vivien Seguy**
Graduate School of Informatics
Kyoto University
vivien.seguy@iip.ist.i.kyoto-u.ac.jp

**Marco Cuturi**
Graduate School of Informatics
Kyoto University
mcuturi@i.kyoto-u.ac.jp

## 1  Pseudo Geodesic Metric

Let us first recall from [Ambrosio et al., 2006, Section 12.4] that the set of geodesics in the Wasserstein space can be identified to some plans having first marginal equal to $\bar{\mu}$,

$$\mathbf{G}(\bar{\mu}) = \left\{ \pi \in P_2(\mathcal{X}^2),\ p_1 \# \pi = \bar{\mu},\ (p_1, p_1 + \varepsilon p_2) \# \pi \text{ optimal for some } \varepsilon > 0 \right\}. \qquad (1)$$

Ambrosio et al. [2006, Definition 12.4.1] defined a metric on $\mathbf{G}(\bar{\mu})$,

$$W_{\bar{\mu}}(\pi_1, \pi_2)^2 = \min \left\{ \int_{\mathcal{X}^3} |x_3 - x_2|^2 d\gamma,\ \gamma \in \Gamma(\pi_1, \pi_2) \right\}, \qquad (2)$$

where $\Gamma(\pi_1, \pi_2) \subset P(X^3)$ is the set of a plans verifying $p_{12} \# \gamma = \pi_1$ and $p_{23} \# \gamma = \pi_2$, with $p_{12}(x_1, x_2, x_3) = (x_1, x_2)$ and $p_{13}(x_1, x_2, x_3) = (x_1, x_3)$. If for example $\pi_2$ is induced by a mapping $T$, namely $\pi_2 = (\mathrm{id} \times T) \# \bar{\mu}$, then this metric has the more simple expression [Ambrosio et al., 2006, page 316],

$$W_{\bar{\mu}}(\pi_1, \pi_2) = \left( \int_{\mathcal{X}^2} \|x_2 - T(x_1)\|_{\mathcal{X}}^2 d\pi_1(x_1, x_2) \right)^{1/2}. \qquad (3)$$

Interestingly, if we look for the $T$ which minimizes $W_{\bar{\mu}}(\pi_1, \pi_2)$ in Equation (4), we get that the solution is unique $\bar{\mu}$-almost surely and is equal to the barycentric projection of $\pi_1$. This can be seen by disintegrating $\pi_1$,

$$W_{\bar{\mu}}(\pi_1, \pi_2)^2 = \int_{\mathcal{X}} \left( \int_{\mathcal{X}} \|x_2 - T(x_1)\|_{\mathcal{X}}^2 d\pi_{1, x_1}(x_2) \right) d\bar{\mu}(x_1). \qquad (4)$$

For each $x_1$ the minimum in the inner integral is achieved indeed for $T(x_1) = \int_{\mathcal{X}} x_2 d\pi_{1, x_1}$, which is the barycentric projection of $\pi_1$. As seen above, if moreover $\pi_1$ is an optimal transport plan, then its barycentric projection is an optimal mapping. This observation motivates definition 1, which introduces a quantification of the difference between two vector fields which can be minimized with the barycentric projection.

**Definition 1** (Geodesic pseudo metric on $L^2(\bar{\mu}, \mathcal{X})$)**.** *Let $u$ and $v$ in $L^2(\bar{\mu}, \mathcal{X})$. Let $\Pi_o^u$ be the set of optimal transport plans between $\bar{\mu}$ and $(\mathrm{id} + u) \# \bar{\mu}$, and $\Pi_o^v$ be the set of optimal transport plans between $\bar{\mu}$ and $(\mathrm{id} + v) \# \bar{\mu}$. We define,*

$$GW_{\bar{\mu}}(u, v) = \inf_{\pi_1 \in \Pi_o^u,\ \pi_2 \in \Pi_o^v} W_{\bar{\mu}} \left( (p_1, p_2 - p_1) \# \pi_1, (p_1, p_2 - p_1) \# \pi_2 \right)$$

*which is the minimal distance between all geodesics starting from $\bar{\mu}$ and going through $(\mathrm{id} + u) \# \bar{\mu}$ at time $t = 1$, and all geodesics starting from $\bar{\mu}$ and going through $(\mathrm{id} + v) \# \bar{\mu}$ at time $t = 1$.*

$GW_{\bar{\mu}}$ does not always satisfy the triangular inequality and is thus not a metric on $L^2(\bar{\mu}, \mathcal{X})$. $GW_{\bar{\mu}}$ becomes a metric when $\Pi_o^w$ contains a unique element for any $w \in L^2(\bar{\mu}, \mathcal{X})$, which is the case for example if $\bar{\mu}$ admits a density. If moreover $\mathrm{id} + u$ and $\mathrm{id} + v$ are optimal mappings, then $\pi_1 = (\mathrm{id} \times (\mathrm{id} + u))\#\bar{\mu}$ and $\pi_2 = (\mathrm{id} \times (\mathrm{id} + v))\#\bar{\mu}$ are the unique optimal plans, and then $GW_{\bar{\mu}}(u,v) = \|v - u\|_{L^2(\bar{\mu}, \mathcal{X})}$. To summarize, these results yield the following Proposition, which motivates the use of the barycentric projection.

**Proposition 1.** *Let $v$ in $L^2(\bar{\mu}, \mathcal{X})$ and $\pi_o^v$ an optimal transport plan between $\bar{\mu}$ and $(\mathrm{id} + v)\#\bar{\mu}$. Assume $\pi_o^v$ is unique and that there exists a solution $w$ to,*

$$w \in \min_{\mathrm{id}+u \in \mathcal{C}_{\bar{\mu}}(\mathcal{X})} GW_{\bar{\mu}}^2(u, v),$$

*such that the optimal transport plan $\pi_o^w$ between $\bar{\mu}$ and $(\mathrm{id} + w)\#\bar{\mu}$ is unique. Then,*

$$w = B((p_1, p_2 - p_1)\#\pi_o^v). \tag{5}$$

*Proof.* Since we assume that the solution $w$ to the minimization problem has the property that there is a unique optimal transport plan between $\pi_o^w$ between $\bar{\mu}$ and $(\mathrm{id}+w)\#\bar{\mu}$, it is equivalent to restrict to the $u$ which also verify this property. The constraint $u \in \mathcal{C}_{\bar{\mu}}(\mathcal{X}) - \mathrm{id}$ means that $(\mathrm{id} \times (\mathrm{id}+u))\#\bar{\mu}$ is an optimal transport plan between $\bar{\mu}$ and $(\mathrm{id} + u)\#\bar{\mu}$, and then $(p_1, p_2 - p_1)\#\pi_o^u = (\mathrm{id} \times u)\#\bar{\mu}$. This leads to,

$$GW_{\bar{\mu}}^2(u,v) = \min_u \int_{\mathcal{X}^2} \|x_2 - u(x_1)\|_{\mathcal{X}}^2 d(p_1, p_2 - p_1)\#\pi_o^v(x_1, x_2),$$

which is minimum if and only if $w$ is the barycentric projection of $(p_1, p_2 - p_1)\#\pi_u^v$ as discussed earlier. □

Although we are not able to compute a solution of Equation (7), the last proposition shows that substituting the $L^2$ norm in Eq. (7) by the pseudo metric defined in definition 1, we have an analytic solution which is simple to obtain through the computation of an optimal transport plan and a barycentric projection. As stated above, this pseudo metric and the $L_{\bar{\mu}}^2$ norm are equal on the subset $\mathcal{C}_{\bar{\mu}}(\mathcal{X}) - \mathrm{id}$ of $L^2(\bar{\mu}, \mathcal{X})$ when $\bar{\mu}$ admits a density.

## 2  MNIST Principal Components per Digits with our approach

1000 images for each of the digits of the MNIST dataset have been sampled. We display below the first three PCs computed with our approach.

Figure 1: Digits 0,1,2,3,4. The three PCs are sampled at times $t_k = k/4, \ k = 0, \ldots, 4$ for each of these digits.

Figure 2: Digits 5,6,7,8,9. The three PCs are sampled at times $t_k = k/4,\ k = 0, \ldots, 4$ for each of these digits.

## 3 MNIST Principal Components per Digits with Wang et al.'s approach (2013)

As above, 1000 images for each of the digits of the MNIST dataset have been sampled. We display below the first three PCs computed using Wang et al.'s approach (2013).

Figure 3: Digits 0,1,2,3,4. The three PCs are sampled at times $t_k = k/4,\ k = 0, \ldots, 4$ for each of these digits.

Figure 4: Digits 5,6,7,8,9. The three PCs are sampled at times $t_k = k/4,\ k = 0, \ldots, 4$ for each of these digits.

# 4   Algorithm Pseudo Code

---

**Algorithm 1** Compute the $(n+1)^{\text{th}}$ generalized geodesic principal component

---

1: **Input:** For $i \leqslant N$ : $X_i \in \mathbb{R}^{d \times n_i}$, $a_i \in \mathbb{R}^{n_i}_+$ in the simplex. $Y \in \mathbb{R}^{d \times p}$, $b \in \mathbb{R}^p_+$ in the simplex. $K \in \mathbb{N}$, gradient step size $\beta > 0$, parameter $\lambda > 0$. $V_1$ and $V_2$ initial random matrices in $\mathbb{R}^{d \times p}$ with small norms.

2: **while** not converged **do**

3:     For all $i$ and $t_k = k/K$, form $M_{Z_{t_k} X_i}$ and solve Eq. (9).

4:     For all $i$, compute the optimal projection time $t_i^\sharp$ and the corresponding optimal plan $P_i^\sharp$.

5:     For all $i$, compute the gradients of $m_i$ as in Eq. (13).

6:     Update

$$V_1 \leftarrow V_1 - \beta \left( \sum_{i=1}^N \nabla_1 m_i^{V_1 V_2} + \lambda \nabla_1 \Omega \right), \;\; V_2 \leftarrow V_2 - \beta \left( \sum_{i=1}^N \nabla_2 m_i^{V_1 V_2} + \lambda \nabla_2 \Omega \right)$$

.

7:     Project $V_1$ and $V_2$ on $\text{span}(V_1^{(1)} + V_2^{(1)}, \cdots, V_1^{(n)} + V_2^{(n)})^\perp$ in the $L_{\bar{\mu}}^2$ sense.

8:     Compute the optimal plans $\hat{P}_1^*$ and $\hat{P}_2^*$ as in Eq. (14).

9:     Update $V_1$ and $V_2$ through Eq. (15):

$$V_1 \leftarrow - \left( (Y - V_1) P_1^{\star T} \text{diag}(b^{-1}) - Y \right), \quad V_2 \leftarrow (Y + V_2) P_2^{\star T} \text{diag}(b^{-1}) - Y.$$

10: **end while**

---