[Reviews · NeurIPS 2015]

Submitted by Assigned_Reviewer_1

- The intro provides no high-level context, and overall, the paper is not at all accessible to someone who doesn't have previous background on optimal transport theory.

- The empirical results are not very motivating.

The authors discuss other methods that have been used to solve the same / similar problems (Boissard, Biot, Wang), yet they don't compare with any of these methods (let alone to standard PCA).

Moreover, as with many unsupervised approaches, its hard to evaluate the quality of the results.

As the authors suggest, it would be nice to perhaps combine this unsupervised approach with a downstream supervised approach to qualitatively assess its performance.

There are several typos in the paper.

Some example include: - abstract: "descend" -> "descent" - intro: Hiblert - Def 1: "the space" -> "be the space" - MNIST: annexe - MNIST: Fig 4 displays - > Fig 5 displays
Summary: This paper presents a new

PCA formulation in Wasserstein Spaces, as well as projected gradient descent approach to approximately solve this problem.

I found this paper very difficult to read, and also had trouble understanding the high-level motivation / significance of the results.

Submitted by Assigned_Reviewer_2

The paper proposes a new algorithm for geodesic principal component analysis (GPCA) in the Wasserstein space. Given Wasserstein barycenter, GPCA wants to find two end points of geodesic which go through the Wasserstein barycenter; in this paper, this problem is computed by projected gradient descent method.

I summarize the strengths (+) and weaknesses (-) of this paper. (+) Designing the objective function (6) is rather straightforward from the previous studies, but solving it involves non-convex optimization whose exact computation algorithm has not been developed. An approximation idea of projection on the optimal mapping set is novel if it is technically correct and has not been applied in this problem. (-) The paper mentions limitations of the existing works on GPCA in the Wasserstein space, but it is unclear how the new algorithm overcomes it. Quick summary of this can be added either in related work subsection or algorithm section. Even if some terms in the objective function (6) are designed to overcome limitations of existing works, actual results may not follow intended properties because of approximation steps to handle several non-convex sub-problems. To check this, I suggest to compare algorithms via numerical experiments.
Summary: The paper proposes a projected gradient descent method for geodesic principal component analysis (GPCA) in the Wasserstein space, which overcomes limitations of the existing works. However, the current draft does not well describe the superiority of the new algorithm.

Submitted by Assigned_Reviewer_3

In this paper, the authors proposed a novel algorithm to compute principal geodesics in the Wasserstein space. Experiments in simulated and real datasets show the promising performance.

Essentially, the PCA is generalized into nonlinear PCA when such estimation is performed under the Wasserstein space, which is a high-dimensional nonlinear space. However, it is not easy to see such nonlinearity is well described through observing several experimental results reported in this paper. Specifically, 1) in Fig, 1, 2 and 3,

I guess these principal Geodesics should be principal components whereas in my opinion, the real principal Geodesics should be principal curves in one-dimensional sense or principal manifolds in two or higher dimensional sense.

Further, since the authors do not explain what the points are,

I have no ideas on how the algorithms run for the data points and whether the results are the best ones. 2) Fig. 4 and Fig. 5 should be used to discover one intrinsic dimension or the first principal Geodesic. Therefore, if the intrinsic dimensions are greater than 3, maybe the algorithms will not work.

I think the authors should discuss the limitation of their proposed algorithms related to the dimension of intrinsic variables. 3) Orthogonality: I agree that the authors are right that in supplementary material, Figure 1 indeed discovered two intrinsic variables: scale and translation. However, for other figures, it is not easy to find such a nonlinear orthogonality. Especially for other images, I have difficult to figure out the "orthogonality" from them. Therefore, I think that the authors should compare their proposed algorithm with not only PCA but also other nonlinear PCA algorithms so that we can see how much the algorithms improve the estimation of principal Geodesics based on different datasets.

Summary: In this paper, the authors proposed a novel algorithm to compute principal geodesics in the Wasserstein space. Experiments in simulated and real datasets show the promising performance.

Submitted by Assigned_Reviewer_4

Detailed comments:

- add a remark/corollary showing that the proposed Wassertein PCs recover the regular Euclidean PCs when considering a Euclidean space

- can you characterize the saddle-points of Problem 6 in the Wasserstein space? What do these saddle-points correspond to in the Wasserstein space? There seems to be an interesting structure here, worth exploring further.

- can you theoretically compute the optimal step-size of the projected gradient descent, depending on unknown constants of the probability measures involved? What drives the scale of this step-size, depending on the probability measures? This is an interesting aspect to interpret.

- what is your numerical criterion for convergence? Please detail and comment on this.
Summary: The paper proposes an algorithm for computing a notion of principal components (PCs) defined with respect to Wasserstein distances (a particular kind of transportation distance between probability measures). The approach uses interesting concepts of Frechet mean, geodesic, tangent spaces/vectors in the Wasserstein space, that allow to rigorously define what is meant by '"Wasserstein principal components".

The authors propose a projected gradient descent algorithm to solve the corresponding non-convex optimization problem, which with multiple initial points leads to accurate estimation of Wasserstein PCs. Visually very convincing experimental results are presented to compute such PCs on a toy dataset and a dataset of shapes.

This is a beautiful paper, proposing a clear-cut and simple algorithm for a long-standing open problem: computing Principal Components in the Wasserstein space. A strong accept.

Author Feedback
Author rebuttal: Thank you all for your valuable reviews.

We have noticed two main criticisms. We will be happy to clarify and correct these misunderstandings in our paper, if given the opportunity:

1 ==== Missing comparison to previous Wasserstein PCA work

R4: "Even if some terms in the objective function (6) are designed to overcome limitations of existing works, [...] To check this, I suggest to compare algorithms via numerical experiments."
R5: "yet they don't compare with any of these methods"

Answer: We cite Bigot'13, Boissard'13 and Wang'13 because we want to give them proper credit for trying to do some form of Wasserstein PCA before we did. However, citing them does not mean that we can or should benchmark our work against theirs:

- Bigot'13 only works for 1D distributions;
- Boissard'13 requires the prior knowledge of a parameterized family of deformations, and is only used in 1D examples;
- Wang'13 do not use Wasserstein barycenters, compute arbitrary curves and not geodesics, and use the usual (very costly) formulation of optimal transport.

Clearly, these works are seminal references, but are not applicable in a machine learning context.

None of these algorithms could run on the examples we consider, due to significant modeling and computational issues. We will clarify further that our approach is the first "clear-cut and simple algorithm for a long-standing open problem" (R.10)

2 ==== Missing comparison to nonlinear PCA

R1: "Essentially, the PCA is generalized into nonlinear PCA"
R1: "should compare their proposed algorithm with not only PCA but also other nonlinear PCA algorithms so that we can see how much the algorithms improve the estimation of principal Geodesics based on different datasets."
R8: "fails to provide convincing evidence that the proposed method is competitive against other nonlinear projection techniques"

Answer: On the space of probability measures P(X), neither PCA nor kernel PCA can produce curves (not to mention geodesics) made of points that all lie in P(X). This is illustrated in Fig. 1 in the supplementary material for standard PCA, or in the references below for kernel-PCA:
http://www.jmlr.org/papers/volume8/guenter07a/guenter07a.pdf (figure 3)
http://www.jmlr.org/papers/volume6/cuturi05a/cuturi05a.pdf (figure 4)
http://colah.github.io/posts/2014-10-Visualizing-MNIST/

We agree with Reviewers 1,8 that both PCA and nonlinear/kernel PCA can produce low dimensional representations of probability measures.

However, this is not our goal: we want to highlight the principal components themselves, i.e. carve out directly geodesics in datasets in their original space, not in some hard-to-interpret embedding space.

We do not know any form of nonlinear PCA that can do this. If Rev.1,8 do, please give us references.

==== Detailed comments:

Rev.1

- "1) in Fig, 1, 2 and 3, [...] the real principal Geodesics should be principal curves in one-dimensional sense"

In each of these Figures, all the orange segments---combined---define indeed ONE single (Wasserstein) geodesic, namely a cloud-of-points interpolating between two clouds-of-points, and not many geosedics. Each of these 3 geodesics is one-dimensional (time t). We believe this remark shows a fundamental misunderstanding of our approach.

- "2) Fig. 4 and Fig. 5 should be used to discover one intrinsic dimension [...]"

Fig. 4 displays the first Principal Component for 10 datasets (one dataset per digit). Other PC's are given in the supplementary material.

- "3) In Fig 6 and 7, it is not easy to see whether the first 3 principal Geodesics are different."

Because all PC's go through the (bary)center, they coincide in their middle. Their extremities are however different.

Rev.10

Many thanks for your comments and your research questions, all worth investigating.

- Indeed, PCA can be recovered as a special case when all measures are single diracs. This will be added.

- At this point we use a naive convergence criterion. Our algorithm stops when the velocity fields v_1 and v_2 have converged, i.e. ||v_1_i-v_1_{i-1}||/||v_1_{i-1}||, and same for v_2, become small enough. We cannot guarantee convergence since the projection step is not done in the L^2 sense. We have always observed convergence experimentally when the step-size (kept fixed) is small enough.

Rev.7

Thanks for your summary, in particular the very useful description that "the method captures shape variations without having to parametrize the shape"

Rev.8

- "Images are not an objective measure of the quality."
We disagree with this criticism. Visual evaluation of unsupervised algorithms is essential (e.g. in clustering, topic modelling, manifold learning), not to mention the whole field of data visualization.